# Trends and drivers of hypoxic thickness and volume in the northern Gulf of Mexico: 1985–2018

**Venkata Rohith Reddy Matli**[1]***, **Daniel Obenour**[1,2]

**1** Center for Geospatial Analytics, North Carolina State University, Raleigh, North Carolina, United States of America, **2** Department of Civil, Construction, & Environmental Engineering, North Carolina State University, Raleigh, North Carolina, United States of America

\* vmatli@ncsu.edu

**Data Availability Statement:** The data is located in the Dryad archive and can be accessed using the DOI: https://doi.org/10.5061/dryad.q2bvq83tw.

**Funding:** This work was funded by NOAA grant NA16NOS4780203 . The funders had no role in

## Abstract

Hypoxia is a major environmental issue plaguing the commercially and ecologically important coastal waters of the northern Gulf of Mexico. Several modeling studies have explored this phenomenon, but primarily focus on the areal extent of the mid-summer hypoxic zone. Research into the variability and drivers of hypoxic volume and thickness is also important in evaluating the seasonal progression of hypoxia and its impact on coastal resources. In this study, we compile data from multiple monitoring programs and develop a geospatial model capable of estimating hypoxic thickness and volume across the summer season. We adopt a space-time geostatistical framework and introduce a rank-based inverse normal transformation to simulate more realistic distributions of hypoxic layer thickness. Our findings indicate that, on average, there is a seasonal lag in peak hypoxic volume and thickness compared to hypoxic area. We assess long-term trends in different hypoxia metrics (area, thickness, and volume), and while most metrics did not exhibit significant trends, mid-summer hypoxic thickness is found to have increased at a rate of 5.9 cm/year ($p < 0.05$) over the past three decades. In addition, spring nitrogen load is found to be the major driver of all hypoxia metrics, when considered along with other riverine inputs and meteorological factors in multiple regression models. Hypoxic volume, which was also often influenced by east-west wind velocities, was found to be more predictable than hypoxic thickness.

## Introduction

Increases in anthropogenic nutrient loading have led to a rise in the number of coastal hypoxic zones (a.k.a. dead zones) across the world [1, 2], threatening coastal ecosystems and fisheries [3, 4]. One of the largest coastal hypoxic zones forms each summer on the Louisiana-Texas shelf of the northern Gulf of Mexico. The primary driver of this dead zone is nitrogen loadings from Mississippi River Basin [5], which stimulates organic production that eventually sinks and results in bottom-water oxygen depletion through microbial decomposition [6]. Physical

study design, data collection and analysis, decision
to publish, or preparation of the manuscript.

**Competing interests:** NO authors have competing
interests.

conditions, such as water column stratification, are also a prerequisite for hypoxia formation
[7, 8], and climate change is expected to exacerbate hypoxia over time [9, 10]. Observations of
Gulf hypoxia date back as early as 1970, but regular monitoring started in the 1980s [11]. The
Mississippi River/Gulf of Mexico Watershed Nutrient Task Force was formed to implement
measures to control nutrient loading and reduce the size of the hypoxic zone to 5000 km$^2$
by the end of 2015, which was later extended to 2035, as the original objective remains unreal-
ized [12].

Over the last several decades, several organizations have conducted monitoring cruises to
assess the extent of hypoxia and its ecological effects. Data from monitoring programs are used
in developing models capable of estimating hypoxic area [13, 14], which are in turn used in cal-
ibrating forecast models [10, 15, 16] and assessing fisheries-related impacts [17]. However,
hypoxic volume and thickness are also potentially important from an ecological and fisheries
perspective [18]. Pelagic and demersal fish are affected by hypoxic volume, as they must move
laterally or vertically to avoid the hypoxic layer [19, 20]. Therefore, estimates of hypoxic vol-
ume, in conjunction with hypoxic area, are needed to characterize the severity of hypoxia and
help inform management efforts.

Hypoxic volume is more commonly considered in other aquatic systems, such as Chesa-
peake Bay [21, 22] and the Baltic Sea [23, 24]. Hypoxic volume in Chesapeake Bay has been
estimated using universal kriging and other spatial interpolation methods [25–27]. In the Bal-
tic Sea, which is approximately 30 times bigger than Chesapeake Bay, coarser spatial and tem-
poral sampling [28, 29] has limited the applicability of geostatistical methods [30].
Additionally, most geostatistical studies are limited to interpolation through kriging. Kriging
provides a smoothed interpolation surface, while conditional realizations (i.e., spatial Monte
Carlo simulations) are needed to reliably characterize overall measures of hypoxia (e.g., area
and volume) and quantify uncertainties [27, 31].

Hypoxic volume in the Gulf of Mexico remains relatively uncharacterized. Obenour et al.
(2013) [13] developed a geostatistical model for estimating hypoxic volume based on dissolved
oxygen (DO) measurements from the Louisiana Universities Marine Consortium (LUMCON)
midsummer shelfwide cruises, and Scavia et al. (2013) [32] incorporated these estimates into a
parsimonious hypoxia forecasting model. However, these models are limited to predicting
hypoxic volume at the time of the LUMCON cruise, typically in late July. Additionally, Obe-
nour et al. (2013) [13] modeled hypoxic fraction (HF, i.e., thickness of hypoxic bottom layer
divided by total water column depth) as a Gaussian random field, which is typical of geostatis-
tical modeling, but does not fully address the highly skewed and bounded nature of actual hyp-
oxic thickness observations. Fennel et al. (2016) [33] developed a hydrodynamic-
biogeochemical model capable of predicting hypoxic volume across the summer, but results
have only been systematically compared to midsummer estimates. Thus, there remains the
need for data-driven estimates of hypoxic volume across the summer season, as intra-seasonal
variability is important for assessing fisheries and ecosystem health [18, 34].

This study advances previous hypoxic volume estimation techniques [13] in two substantial
ways. First, it applies a space-time geostatistical framework [14] to observations of hypoxic
layer thickness from multiple monitoring organizations. By accounting for spatial and tempo-
ral trends and correlations, this approach enables interpolation of hypoxic thickness across the
summer season. Second, it investigates a rank-based inverse-normal transformation to address
the skewed and bounded data distribution of HF. Through conditional realizations, the model
is used to probabilistically estimate the volume and thickness of hypoxia across the Gulf hyp-
oxic region from May through September (1985–2018). We also explore trends in area, vol-
ume, and thickness of hypoxia. This includes the development of regression models to
compare the predictability and key drivers of the different hypoxia metrics.

## Methods

### Data and domain

DO data are derived from the monitoring programs described by Matli et al. (2018; 2020) [14, 34], as available through NOAA's National Centers for Environmental Information (NCEI). Compared to Matli et al. (2020) [34], we add monitoring cruises conducted by LUM-CON (2015, 2017, and 2018) and the Southeast Area Monitoring and Assessment Program (SEAMAP) (2018, 2019) (Table 1 in S1 Text). The total dataset comprises 7939 sampling events collected from 186 monitoring cruises. While Matli et al. (2020) focused only on bottom-water dissolved oxygen (BWDO), this study requires the entire vertical DO profile to determine the bottom-layer hypoxic thickness, and events that do not span the entire water column are filtered out (4.5% of sampling events) [34]. Hypoxia is primarily a bottom-water phenomenon, and observations with HF>0.75 (only nine events) may result from measurement error or ephemeral biological and hydrodynamic conditions that result in high levels of near-surface respiration [6, 35]. Thus, we cap these "outlier" observations at 0.75. The overall dataset includes 2823 observations of HF greater than zero (histograms provided in Fig 1 in S1 Text). Our study area is bounded by 94.605–89.512˚ W longitude, 28.219–29.717˚ N latitude, and 3–75 m depths, and divided into a 5x5 km grid for estimation purposes.

The DO profiles were collected using handheld or rosette-mounted samplers [13]. In general, rosette-mounted samples did not sample the sea floor to avoid damaging the sampling frame. A correction factor was applied to account for this bias, similar to Obenour et al. (2013) [13], as described in S1 Text.

### Model formulation

Geostatistical models perform optimally when the response variable can be represented as a stationary Gaussian process [31, 36]. Obenour et al. (2013) [13] proposed modeling HF, rather than thickness, to partially address this issue (hypoxic thickness tends to increase with depth). However, HF values remain right skewed (Fig 1 in S1 Text) and so in this study we transform HF using a rank-based inverse-normal transformation (INT) via Blom's function [37]. Rank-based INT converts a continuous variable (with any underlying distribution) to a normal distribution by applying a quantile function to observations ranked in ascending order. To efficiently back-transform to the original scale, we apply a piecewise quadratic regression. The coefficients and breakpoints of this regression are estimated by minimizing the mean square error when fit to observations at equal intervals along the transformed scale (Fig 1).

A space-time geostatistical model can be conceptualized as follows using transformed HF ($HF_t$) as the response variable (Eq (1)):

$$HF_t = \mu + \eta + \varepsilon \qquad (1)$$

where $\mu$ is the deterministic component (trend in mean) and $\eta$ and $\varepsilon$ are stochastic components. Here, $\mu$ is a linear model with trend variables (covariates) and variable intercepts (categorical variable), with the latter accounting for interannual variability. Covariates include linear or quadratic trends with northing (N), easting (E), depth (D), day-of-year (T), and BWDO. Variable selection using a geostatistical adaptation of the Bayesian Information Criterion (BIC) is performed to avoid over-parameterization [13]. The full system of linear equations for the space-time geostatistical model are provided in Matli et al. (2018) [14].

The variability around the deterministic component is resolved into correlated and uncorrelated stochasticity. Uncorrelated stochasticity ($\varepsilon$) represents environmental microvariability or sampling error. Correlated stochasticity ($\eta$) accounts for transient spatial and temporal

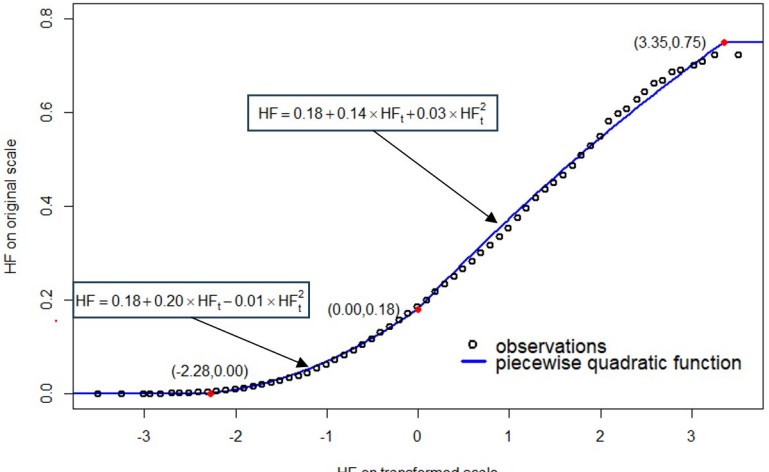

**Fig 1. Original versus inverse-normal transformed (INT) values of hypoxic fraction (HF versus HF$_t$).** The piecewise-quadratic function is used for back-transformation. Solid red circles indicate the breakpoints in this function.

patterns in the data. The covariance $Q$ between sampling events $i$ and $j$, separated by distance $s_{ij}$ and time lag $t_{ij}$, is modeled using a non-separable, space-time covariance function with exponential form (Eq (2)) [14]:

$$Q\left(s_{ij}, t_{ij}\right) = \begin{cases} \sigma_\varepsilon^2 + \sigma_\eta^2, & s_{ij} = t_{ij} = 0 \\ \sigma_\eta^2 \times \exp\left(-\sqrt{\dfrac{s_{ij}^2}{a^2} + \dfrac{t_{ij}^2}{b^2}}\right), & s_{ij} \text{ or } t_{ij} > 0 \end{cases} \tag{2}$$

where $\sigma_\varepsilon^2$ represents the correlated variance (partial sill) and $\sigma_\eta^2$ represents uncorrelated variance (nugget) in HF$_t$. Parameters $a$ (km) and $b$ (days) correspond to approximately one third of the spatial and temporal correlation ranges, respectively. Note that north-south distances are scaled by an anisotropy ratio, $\alpha$ (see Results). Covariance function parameters are estimated using restricted maximum likelihood [38] to account for the selected trend variables, as outlined in Matli et al. (2018) [14].

The fitted covariance function is used to generate unconditional realizations of HF$_t$, which are then conditioned to the observed data and covariate trends [14, 31]. These conditional realizations (CRs) account for uncertainty in the covariate trends as well as the stochastic components of the model. CRs of BWDO [14] are also required, as BWDO is one of the deterministic covariates for HF$_t$. Moreover, simulations of hypoxic thickness are only appropriate at locations that experience hypoxia (BWDO < 2 mg/L). Thus, we adopt the two-step procedure developed by Obenour et al. (2013) [13], where for each CR of BWDO, we first identify the locations that are hypoxic and then perform a CR of HF$_t$ over these locations. This procedure is repeated 1000 times, and the simulations of HF$_t$ are back-transformed to HF and then multiplied by depth to get hypoxic thickness. Next, thickness values are aggregated across the estimation grid to determine hypoxic volume. Results from the 1000 CRs are used to calculate the mean and 95% confidence intervals of these estimates. This process is repeated at a 3-day interval from May to September for 1985–2019. Estimates of hypoxic area, volume, and thickness are highlighted for the LUMCON midsummer cruises (MC) and as summer averages (SA, June-August). To calculate SA hypoxia metrics, we used simulations of area, volume, and

thickness for all estimation dates between June and August to determine the mean. For 95% confidence intervals of SA estimates, we summed variances using the properties of correlated random variables [39], where correlations among 3-day interval estimates for each year were calculated using lagged correlation coefficients. The geostatistical model was developed in MATLAB, and R was used for processing and visualizing model results [40, 41].

## Model verification

The INT is not commonly used in geostatistical modeling and has never (to our knowledge) been used in hypoxia modeling. To investigate the efficacy of this transformation, we compare its ability to produce realistic simulations of hypoxia on four different quadrants of the study area, divided by the 20-m isobath into shallow and deep regions and by the outlet of Atchafalaya River into east and west sections. For each quadrant, we compare the distribution of simulated hypoxic thickness (with and without the INT) to that of the observations.

## Regression analysis of hypoxia metrics

The MC and SA estimates are used as response variables in multiple linear regressions with available hydro-meteorological drivers. For SA, we only use estimates from 1992–2015, which is a continuous period that includes multiple cruises within each year, but with varying geographic footprint, consistent with Del Giudice et al. 2020 [9]. Candidate predictor variables were motivated by previous studies [9] and include aggregations of nutrient loading over winter and spring seasons [42], river flows in summer [43], east-west wind velocities over spring and summer seasons, and wind stress around the time of prediction [44]. For MC models, "spring" refers to April-June and "summer" refers to July, as the LUMCON MC typically takes place in late July. For SA models, "spring" generously refers to April-July and "summer" refers to June-August. For both MC and SA, winter refers to November-March. Finally, wind stress (represented by wind speed squared) was determined as a 14-day weighted average (weights linearly declining backward in time from the cruise mid date) when predicting MC hypoxia [45], or as a June-August average when predicting SA hypoxia. We also tested for interactions between westerlies and nitrogen loads at spring and summer seasonal scales.

We developed regressions for hypoxia metrics aggregated across the entire Louisiana-Texas shelf study area, and across east and west shelf sections, divided at the Atchafalaya River outfall. For east and west sections, we used inputs from the Mississippi and Atchafalaya rivers, respectively, and for shelfwide metrics we used their summation. Conventional BIC was used in an exhaustive search for variable selection to prevent overfitting [46]. We used mean hypoxia estimates in the variable selection process. However, we used 100 samples of the hypoxia metrics (i.e., from the CRs) to determine the average variance explained ($R^2$) in each case.

## Results and discussion

### Geospatial trends and stochasticity

The BIC-selected trend variables (Table 1) and annual intercepts (Table 2 in S1 Text) explain 30.1% of the spatio-temporal variance in transformed HF. The trend with northing indicates that for every 100 km further north, HF (back-transformed) decreases by 0.08 (relative to a baseline, average HF of 0.21). The quadratic trend with depth indicates that HF is at a maximum at 14 m of depth (all else being equal), and the quadratic trend with BWDO indicates that HF is greatest when BWDO is zero and gradually declines as BWDO increases (Fig 2 in S1 Text). The combined trends indicate that the highest HFs are typically at mid-depth regions on the eastern shelf (Fig 3 in S1 Text). Annual intercepts account for additional year-to-year

**Table 1. Trend coefficients ($\beta$) and associated standard errors ($\sigma_\beta$) of BIC selected variables for $HF_t$ and BWDO model.**

| Covariate | $HF_t$ (unitless) | | BWDO (mg/L) | |
|---|---|---|---|---|
| | $\beta$ | $\sigma_\beta$ | $\beta$ | $\sigma_\beta$ |
| E (km) | - - | - - | -0.0044 | 0.0004 |
| N (km) | -0.0059 | 0.0011 | -0.0086 | 0.0015 |
| D (m) | 0.0074 | 0.0072 | -0.2620 | 0.0109 |
| $D^2$ ($m^2$) | -0.0003 | 0.0001 | 0.0055 | 0.0002 |
| T (day) | - - | - - | 0.0042 | 0.0026 |
| $T^2$ ($day^2$) | - - | - - | 0.0003 | 0.0000 |
| $BWDO^2$ ($mg^2/L^2$) | -0.4017 | 0.0145 | n.a. | n.a. |

*Note*: Units of trend coefficients ($\beta$) of $HF_t$ model are 1/unit of covariate, and BWDO model are mg/L per unit of covariate.

variability in $HF_t$ (beyond the effect of BWDO) and have a range of 1.51 $HF_t$ from the lowest year (1988) to the highest year (2016), indicating a range of about 0.26 for untransformed HF.

Trends in the BWDO model also influence patterns in hypoxic layer thickness. BWDO trends are the same as those described in Matli et al. (2018) [14], though the coefficients (Table 1) changed slightly (less than 10%) due to the inclusion of new data for 2017–2019. Notably, the BWDO trend model includes a quadratic, intra-annual temporal trend, indicating that hypoxia typically peaks in mid-July. There is also a negative trend with easting indicating BWDO decreases closer to Mississippi River outflow (Fig 4 in S1 Text).

The stochastic components of the geostatistical model explain $HF_t$ variability not captured by the deterministic trends. About half (50.4%) of this remaining variability is spatially correlated within ranges of 69 and 43 km along the east-west and north-south axes (accounting for anisotropy) and temporally correlated within a lag of 12 days. The other half of the stochasticity (49.6%) can be attributed to sampling error and environmental microvariability (nugget). This is a relatively large nugget compared to the BWDO model (Table 3 in S1 Text) and suggests that estimates of hypoxic volume will be more uncertain than area.

## Conditional realizations and hypoxia metrics

The mean of simulated hypoxic thickness values (from CRs) can be mapped across the study area. Hypoxia in 2008 was relatively severe, especially in July (Fig 2, left panels), when compared to the long-term average (Fig 2, right panels). Also shown are the regions where there is at least a 50% probability of hypoxia occurring. Across our multi-decadal study period, approximately 12% of estimation locations have a greater than 50% chance of experiencing hypoxia in July and August, compared to only 8% in June. The most consistent and severe hypoxia is on the eastern shelf, closer to the Mississippi River outfall.

Using the CRs, we determine mean and 95% confidence intervals for shelfwide hypoxic area and volume across each summer from 1985–2019 (Figs 3 and 5–14 in S1 Text). In 2003, for example, hypoxia was unusually mild, with a peak volume only about a third that of 2008. The years 2005 and 2008 highlight the inter- and intra-annual variability in hypoxia. There was a relatively late peak in hypoxia in 2005, while in 2008 hypoxia likely peaked in early July. The average thickness of the hypoxic bottom-layer was typically around 3 to 5 meters, with relatively low values in 2003. The 95% estimation intervals are fairly wide, in general, but are

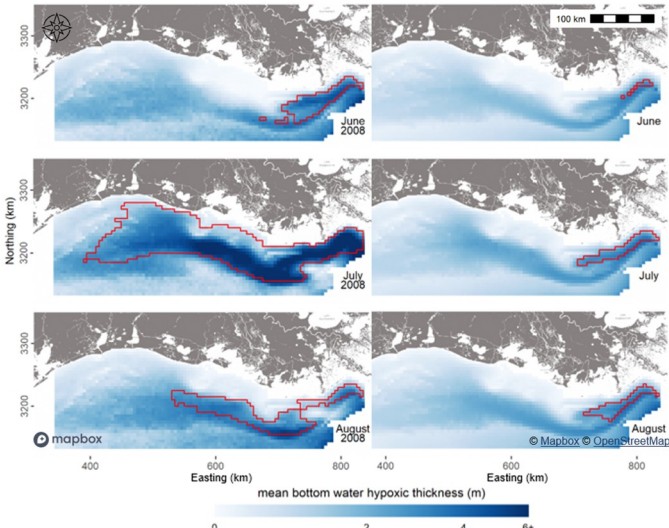

**Fig 2. Mean of simulated bottom water hypoxic thickness on 15 June, 15 July, and 15 August in the relatively severe year of 2008 (left) and across years (right).** The regions outlined in red indicate a greater than 50% probability of being hypoxic (based on the BWDO model) [47, 48].

more constrained in periods of relatively intensive sampling and when hypoxia is less severe (Fig 3).

Throughout the study period, hypoxic area and volume exhibited similar intra-seasonal patterns (Fig 3). Based on results for years with at least two shelfwide cruises, hypoxic area reached its annual estimated peak in May, June, July, August, and September in 0, 3, 15, 6, and 0 years, respectively. For volume, the corresponding results are 0, 2, 11, 10, and 1 years, respectively, indicating that peak volume tends to lag peak area, at least on average. Of course, the results for any given date have substantial uncertainty (see 95% intervals, Fig 3), but taken

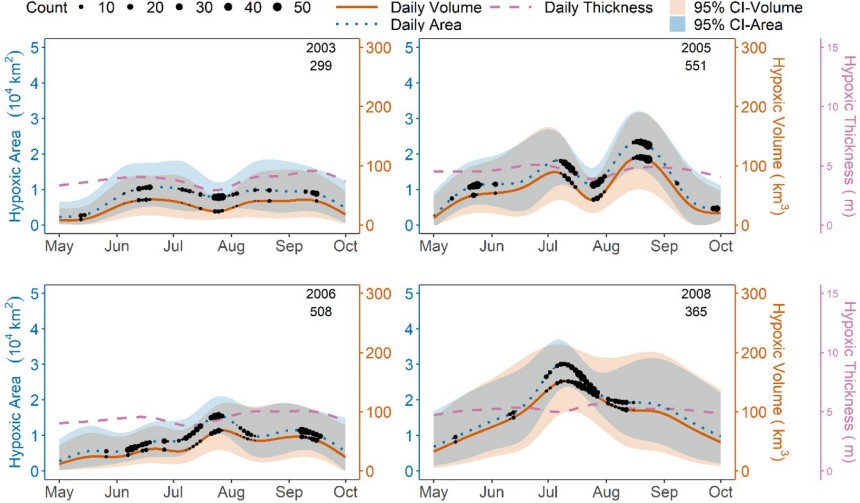

**Fig 3. Shelfwide hypoxic area, volume, and thickness versus time, with 95% confidence intervals for area and volume estimates.** Number of observations through time are overlaid as black dots and the total number is reported below the date. Note that thickness values are averaged over hypoxic locations only.

together, these results suggest both patterns as well as aberrations in the seasonal progression of hypoxic area and volume.

Previous process-based modeling research has also estimated the seasonal progression of hypoxia in specific years (e.g., 2005 and 2006; [33]), which can be compared to our results. In general, process-based simulations suggest more small-scale temporal variability than our estimates. In the geostatistical model, temporal microvariability is reflected in the wide 95% estimation intervals. Simulations of volume from the Regional Ocean Modelling System (ROMS) and Finite Volume Community Ocean Model (FVCOM) indicate that hypoxic volume peaked in August and September of 2005 and 2006, respectively. For 2005, the estimates of peak volume from FVCOM were similar to our estimates (within 10%), but predictions from ROMS were approximately 40% higher, though within our 95% intervals. For August and September 2006, predictions from both FVCOM and ROMS were at least 100% higher than our estimates and outside of our 95% intervals. Importantly, our geostatistical estimates are informed by monitoring cruises conducted in August and September of 2006 (Fig 3), reducing their uncertainty. For another process-based model, the Navy Coastal Ocean Model (NCOM), peak hypoxic volume simulations in 2005 [33] were at least 50% lower than our estimates and outside our 95% intervals. NCOM simulations for 2006, however, compare well to estimates from our model with a similar peak volume and temporal evolution. Considering these examples, there is often substantial disagreement between hypoxic volume estimates from different process-based models, and our geostatistical estimates provide an important line of evidence that could help guide future improvements to hydrodynamic-biogeochemical modeling.

## Model verification

Simulations (i.e., CRs) of hypoxic thickness can be compared to the original observations to help ensure that they are realistic (Fig 4). At the same time, we do not expect simulations to

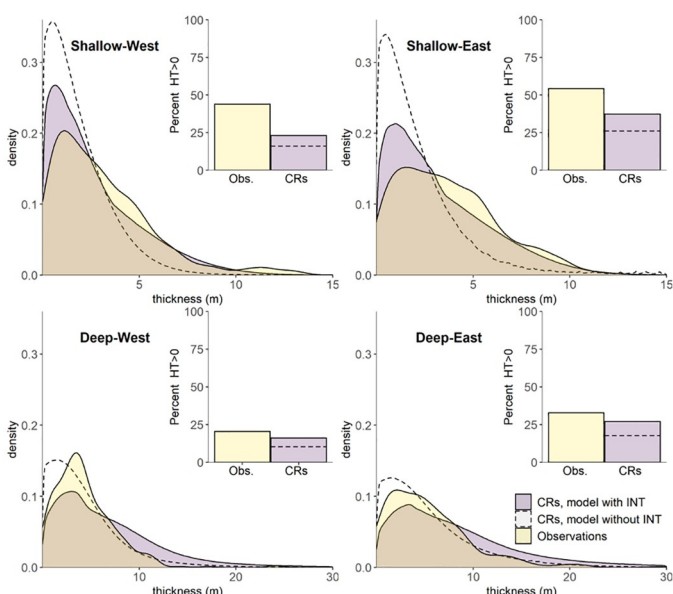

**Fig 4. Probability density distributions of observed hypoxic thickness and geostatistical simulations (i.e., conditional realizations, CRs) of hypoxic thickness across four sections of our study area (June-August).** The deep-shallow divide is the 20-m isobath. The inset bar plots show the percent of simulations results in HT>0 versus corresponding percent of observations. Dashed lines show results for a model developed without using the inverse normal transformation (INT).

perfectly mirror the observational distributions, as many of the monitoring cruises are biased toward areas and times of intense hypoxia (e.g., LUMCON mid-summer cruises specifically target the hypoxic zone) or specific regions of the shelf (Louisiana Department of Wildlife and Fisheries targets near-shore, eastern regions). In general, 20–55% of observations tend to be hypoxic depending on the region of the shelf (see bar charts of HT>0 in Fig 4). The intensity of hypoxia decreases to the west and further offshore, away from the influence of the Mississippi and Atchafalaya River outfalls, as expected based on previous studies [49, 50]. CRs tend to have a somewhat lower percent occurrence of HT>0 (~15–35%, Fig 4) than observations, consistent with expectations, since the model is not biased toward times and locations of intense hypoxia.

The distribution of HT varies substantially across different portions of the shelf (Fig 4, Tables 4, 5 in S1 Text). In the shallow regions (<20 m), most observations and simulations of HT are less than 5 m, while in the deeper regions, around half of HT values exceed 5 m. Comparisons between east and west suggest fewer differences, although the east section appears to have more frequent occurrences of hypoxic thicknesses exceeding 5 m. While all results show substantial right-skew, the CRs in the deep regions exhibit more variance than the observations, while in the shallow regions they exhibit less variance.

For comparison, we also present results based on a geostatistical model developed without the INT of HF. Without this transformation, many HF values are simulated as negative (and treated as zero) even though BWDO is simulated as hypoxic (below 2 mg/L). This is apparent in the substantial reduction in percent HT>0 for the model without the INT (dashed lines in Fig 4 bar plots). With the transformation, simulations of zero HT in hypoxic waters are avoided, and the occurrence rate of HT>0 is more realistic. Moreover, omitting the INT produces distributions of hypoxic thickness with a negative bias and minimal variance, particularly in shallower shelf regions (Fig 4, dashed curves). On the other hand, in the deeper sections, the INT tends to result in distributions with a positive bias and more variance, such that there is not an improvement in the distributions of hypoxic thickness for deeper locations.

Overall, the model with INT provides more realistic simulations of HF and thickness relative to the standard Gaussian geostatistical model. Without the INT, the model appears to under-simulate occurrences of HT>0 across the entire study area, and it has a negative bias in the shallow shelf sections, where hypoxia is most common. Over the entire shelf, the model without INT under-simulates occurrences of HT>0 by 61% relative to the observations, and it under-simulates mean HT (across the entire study area) by 61% relative to observations (0.57 m vs. 1.48 m). At the same time, the model with INT simulates HT>0 at a rate of 30% across June-August, and results in an overall average hypoxic thickness of 1.03 m, more comparable to the observations (Tables 4, 5 in S1 Text).

## Comparison of hypoxic area, volume, and thickness

To help characterize interannual variability, we present the June-August summer average (SA) of hypoxic area, volume, and thickness (Fig 5). We also report these hypoxia metrics at the time of the LUMCON midsummer cruise (MC) for consistency with previous studies [13]. A comparison of MC results from this study with those from Obenour et al. 2013 is provided in S1 Text (Section 8). For 2016 and 2019, it is important to note that data were only available from SEAMAP, which conducts monitoring in late June and early July over limited portions of the study area, often before hypoxia peaks. Geostatistical estimates of SA hypoxia for these years have excessive uncertainty and are unusually large (Fig 5), indicating that the geostatistical model is unable to constrain estimation uncertainties without the benefit of more detailed

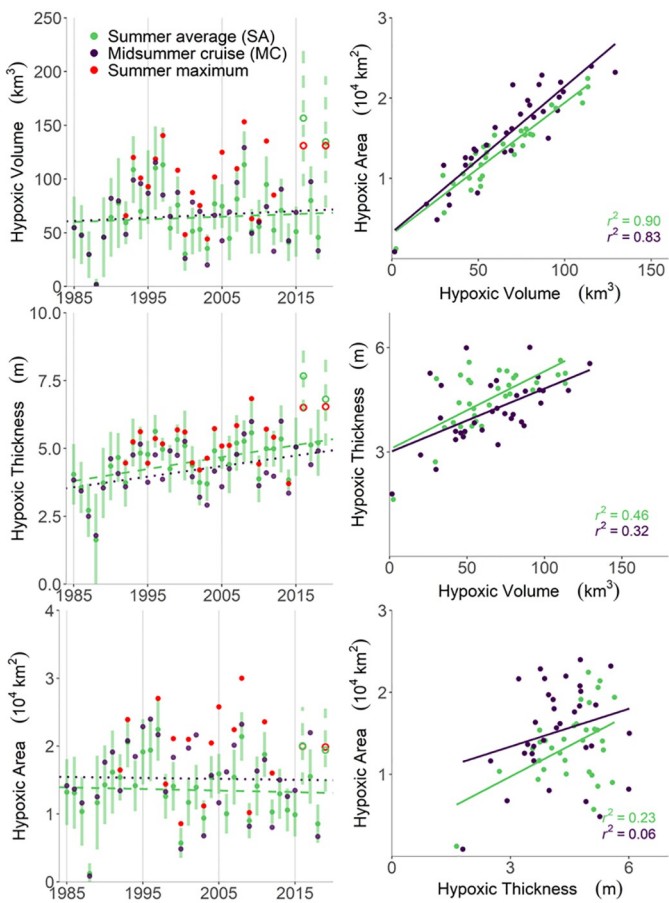

**Fig 5. Estimates of hypoxic volume, thickness, and area for summer average conditions (SA), LUMCON midsummer cruise (MC), and maximum shelfwide cruise (when different from MC), along with trend lines for SA and MC (left panels); correlations among estimates of area, volume, and thickness (right panels).** Note that 2016 and 2019 estimates (hollow circles) are based on limited data and not included in right panels and subsequent analyses.

(e.g., LUMCON) cruises collected closer to the period of peak hypoxia (late July and August). Thus, 2016 and 2019 are omitted from the following correlation and trend analyses.

We compare correlations among the various hypoxia metrics generated in this study (Fig 5). Area and volume are most correlated with each other, with $r^2$ values of 0.83 and 0.90 for MC and SA estimates, respectively. For thickness and volume, correlations drop to 0.32 and 0.46 for MC and SA, respectively. Finally, hypoxic area and thickness were least correlated, with $r^2$ values of 0.06 and 0.23 for MC and SA estimates, respectively, suggesting these two metrics are largely independent. The correlations among SA estimates are consistently higher than the correlations between MC, potentially because MC estimates reflect short-term meteorological variability that influences the structure of hypoxia, while SA estimates tend to average out these fluctuations.

MC and SA estimates were also analyzed for significant long-term temporal trends. To account for estimation uncertainty, we took 1000 samples from the distributions of the hypoxia estimates (CRs). The trends from 1000 samples and the associated level of significance (p-value from two-sided test) are averaged to determine the overall significance level of trends in MC hypoxia metrics. Consistent with results from Matli et al. (2018) [14], there is no significant temporal trend in estimates of hypoxic area or volume (p-val>0.1). However, MC

thickness estimates appear to be increasing at a rate of 5.9 cm/yr (p-val<0.05), while SA thickness is increasing at a rate of 4.4 cm/yr (p-val<0.05). These results are generally consistent with Obenour et al. (2013) [13], who found a significant trend in hypoxic layer thickness (6.9 cm/yr) modeling only MC data through 2011.

In addition to LUMCON MC estimates, there are estimates from other shelfwide monitoring programs in 26 of the 33 years considered. The estimate associated with one (or more) of these other cruises was higher than the MC estimate in 11, 18, and 20 years for area, volume, and thickness, respectively (Fig 5, red dots). In these years, the MC estimate was exceeded, on average, by 46%, 52%, and 25% for area, volume, and thickness, respectively. This, along with the substantial intra-seasonal variability in hypoxia (e.g., Fig 3) demonstrate that MC estimates are not always representative of hypoxic severity for a given year.

## Drivers of hypoxic area, volume, and thickness

Much of the research on Gulf hypoxia focuses on how riverine inputs influence hypoxic area, as this is the metric that traditionally drives watershed management policies [15]. However, hypoxic volume and thickness also play an important ecological role and remain relatively unexplored in the Gulf [18, 51]. Here, we provide a comparison of the predictability of hypoxic area, volume, and thickness based on various loading and meteorological inputs that have been considered in previous modeling studies [9].

Results show that inter-annual variability in hypoxia metrics (1992–2015) can be explained to varying degrees across different shelf sections, with $R^2$ values ranging from 0.07–0.56 (Table 2). In general, volume estimates are less predictable than area estimates (26% lower $R^2$,

**Table 2. Regression fit ($R^2$) and coefficients for BIC-selected variables for models of hypoxic area (A), volume (V), and average thickness (T) across the entire shelf (i.e., shelfwide, SW), and east (E) and west (W) of the Atchafalaya River outfall.** Models were developed for both midsummer LUMCON cruise (MC) and summer-average (SA) estimates.

| Metric | Region | Period | $R^2$ | Candidate Variables | | | | | | |
|---|---|---|---|---|---|---|---|---|---|---|
| | | | | year | spring N loads (Gg/mo) | summer westerly (m/s) | spring westerly (m/s) | winter N loads (Gg/mo) | windspeed squared ($m^2/s^2$) | summer flows ($m^3/s$) |
| A ($10^3$ km$^2$) | SW | MC | 0.46 | | 0.085 | | -3.22 | | | |
| | | SA | 0.41 | | 0.047 | | | 0.09 | | |
| | E | MC | 0.56 | | 0.041 | 0.57 | | | -0.07 | |
| | | SA | 0.39 | | | | | 0.03 | | 0.11 |
| | W | MC | 0.47 | | 0.194 | -1.98 | -3.20 | | | |
| | | SA | 0.43 | | 0.143 | -2.68 | | 0.31 | | |
| V (km$^3$) | SW | MC | 0.36 | | 0.601 | | | | | |
| | | SA | 0.26 | | 0.437 | | | | | |
| | E | MC | 0.55 | | 0.571* | 3.27 | -13.8* | | -0.34 | |
| | | SA | 0.24 | | 0.202 | | | | | |
| | W | MC | 0.34 | | 0.979 | -6.65 | -15.2 | | | |
| | | SA | 0.19 | | 0.971 | | | | | |
| T (m) | SW | MC | 0.28 | 0.04 | 0.013 | 0.28 | | | | |
| | | SA | 0.10 | | 0.007 | | | | | |
| | E | MC | 0.21 | | | | | | -0.04 | 0.10 |
| | | SA | 0.09 | | 0.011 | | | | | |
| | W | MC | 0.14 | 0.05 | | | -0.57 | | | |
| | | SA | 0.07 | | 0.020 | | | | | |

[a] Asterisks (*) denotes an interaction between spring nitrogen (N) loads and spring westerly wind velocity for E MC volume (with a coefficient of 30.18).

[b] Model intercepts provided in Section 4 of S1 Text.

on average), and thickness estimates are less predictable than both area (65% lower $R^2$, on average) and volume estimates. Across all three hypoxia metrics, MC estimates are more predictable than SA estimates. All selected predictor variables are significant at a 95% confidence level (p-val<0.05).

Consistent with most research conducted in the Gulf region [45, 52], spring nitrogen load from the Mississippi River Basin is found to be the primary driver of hypoxic area, volume, and thickness. Winter nitrogen load is also found to be a significant predictor of SA hypoxic area estimates, consistent with Del Giudice et al. (2020) [9], but it is not selected in models for volume and thickness. Summer flows, which are expected to increase the density gradient and slow the reaeration of bottom waters, were selected as significant predictors in just two of the models. Year is selected in two models of hypoxic thickness, confirming a positive temporal trend that is independent of nutrient loading.

Wind velocities are also expected to influence the formation, propagation, and persistence of hypoxia. The associations between east-west wind velocities (westerlies are positive) and hypoxia metrics remained consistent within the different shelf sections. Wind variables are mostly selected in models for MC estimates but appear to have limited relevance for predicting SA estimates, perhaps because meteorological variability tends to be averaged out over longer time periods. Wind shear (weighted wind speed squared over two weeks preceding the mid date of the MC) was also occasionally selected, likely because it disrupts stratification, allowing for reaeration of the water column [53].

Summer winds have varied influence depending on the section being considered. For hypoxic area and volume, summer westerlies promote hypoxia on the eastern section, but tend to reduce it on the western section. This is consistent with previous research [54, 55] showing that upwelling (westerly) winds in June-July lead to a higher likelihood of hypoxia on the eastern shelf but reduce hypoxia on the western shelf. For thickness, however, summer winds are only predictive of shelfwide estimates (not east or west section estimates), indicating they lead to thicker hypoxic layers. This may be a result of a rise in the pycnocline along the coast in response to upwelling circulation patterns [56].

In the spring, easterly wind patterns (i.e., negative east-west winds) prevail on the Gulf shelf. Strong spring easterlies are associated with increased MC hypoxia estimates on the western shelf (and also the entire shelf, for hypoxic area). Additionally, an interaction between spring nitrogen loads and spring winds is selected for MC hypoxic volume on the eastern section. Overall, this interaction indicates that increasing easterly winds reduce hypoxia on the eastern shelf, opposite to the effect for the western shelf (Fig 6). The interaction also suggests that nutrient loading becomes more influential on the eastern shelf when easterlies are weak (Fig 6, varying slopes). Weak easterlies likely result in an increased availability of nutrients in

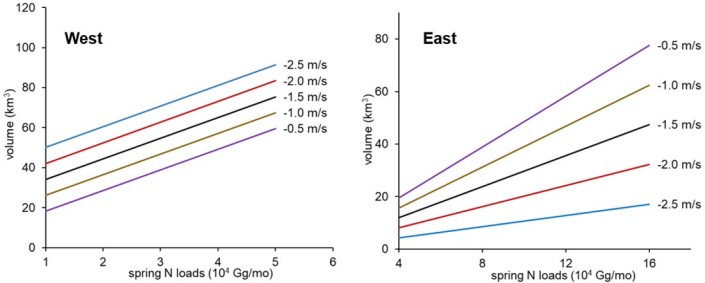

**Fig 6. Influence of interaction between spring nitrogen (N) load and spring east-west wind velocity (different line colors) for the models of MC hypoxic volume.**

the delta region between Mississippi-Atchafalaya River outlets [57]. Other studies conducted in the region found similar effects of east-west winds on formation of hypoxia [45, 54, 58].

## Limitations

The geostatistical model developed in the current study uses spatio-temporal data collected at irregular cadence and limited spatial coverage to provide continuous estimates of hypoxic volume and thickness across the summer in the Gulf of Mexico region. While these estimates provide new insights to the propagation and distribution of hypoxia in the region, the model has certain limitations. The accuracy of geostatistical models is highly dependent on the quality and availability of data at regular intervals. Data availability at irregular cadence limits the model's ability to constrain uncertainty in estimated values during early and late summer months with minimal monitoring efforts. Additionally, the geostatistical model does not consider hydrodynamic-meteorological factors that influence hypoxia dynamics. Future work could integrate dynamic covariates (e.g., coastal winds) directly within the geostatistical model to help capture intra-seasonal variability, as in the fusion approach employed in Matli et al. 2020 for hypoxic area [34]. Furthermore, the regression analysis on hypoxia metrics has limited scope and could benefit from considering additional riverine and meteorological inputs.

## Conclusions

Through the current study, we develop a geostatistical model capable of estimating thickness and volume of coastal hypoxia across the summer based on the limited observations of various monitoring programs. Transforming HF values to a more Gaussian distribution (i.e., using INT) within the geostatistical model results in more realistic simulations of hypoxic layer thickness across our study area, based on comparisons with observational distributions. However, years with no midsummer cruises (2016, 2019) result in excessive estimation uncertainty, highlighting the need for reliable monitoring mid-to-late summer, when hypoxia is typically most severe. At the same time, estimates for years with multiple cruises are more constrained and suggest the need for improvements to hydrodynamic-biogeochemical models, like ROMS, FVCOM, and NCOM, which may not always provide realistic simulations, especially toward the end of the summer season. Overall, the intra-seasonal variability and uncertainty in hypoxia estimates reinforce the need for frequent hypoxia monitoring [50].

Long-term trend analyses corroborate the results from Matli et al. (2018) [14] and Obenour et al. (2018) [13], indicating no significant trends in hypoxic area or volume since 1985, but suggest an increase in hypoxic layer thickness. This temporal trend also appears in some multiple linear regressions for hypoxic thickness, indicating it exists independent of seasonal nutrient loading. Assessing the cause of this increasing trend is beyond the scope of this study, but will hopefully stimulates further research, as it may be related to changes in physical properties (i.e., changes in pycnocline depth) or changes in the nature of oxygen demands over time [53].

Multiple linear regression analysis based on common hypoxia predictor variables, finds spring nitrogen loads from the Mississippi River Basin to be the primary driver of hypoxia across all shelf regions and metrics (area, volume, thickness). Additionally, the analysis shows that volume and (especially) thickness are harder to predict than hypoxic area. These results, along with low correlations between hypoxic area and thickness over time, emphasize the need for considering multiple metrics in hypoxia management. Most research in the Gulf region focuses on evaluating the effects of changing BWDO conditions and hypoxic area on fish and shrimp catch rates. The availability of spatially and temporally resolved estimates of hypoxic thickness from this study can further enhance our understanding of hypoxia effects on aquatic life in the Gulf.

## Supporting information

**S1 Text. Consisting of 10 sections with 14 figures and 10 tables.**
(DOCX)

## Author Contributions

**Conceptualization:** Venkata Rohith Reddy Matli, Daniel Obenour.

**Data curation:** Venkata Rohith Reddy Matli.

**Formal analysis:** Venkata Rohith Reddy Matli.

**Methodology:** Venkata Rohith Reddy Matli, Daniel Obenour.

**Software:** Venkata Rohith Reddy Matli.

**Supervision:** Daniel Obenour.

**Visualization:** Venkata Rohith Reddy Matli.

**Writing – original draft:** Venkata Rohith Reddy Matli.

**Writing – review & editing:** Daniel Obenour.

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
