## [Decision Letter · Decision Letter 0]

3 Jun 2024

PONE-D-24-14571Trends and drivers of hypoxic thickness and volume in Northern Gulf of Mexico: 1985-2018PLOS ONE

Dear Dr. Matli,

Thank you for submitting your manuscript to PLOS ONE. After careful consideration, we feel that it has merit but does not fully meet PLOS ONE’s publication criteria as it currently stands. Therefore, we invite you to submit a revised version of the manuscript that addresses the points raised during the review process.

We look forward to receiving your revised manuscript.

Kind regards,

Abdul Azeez Pokkathappada, Ph.D.

Academic Editor

PLOS ONE

“This work was funded by NOAA grant NA16NOS4780203.”

“This work was funded by NOAA grant NA16NOS4780203. This is NGOMEX contribution ###.”

“This work was funded by NOAA grant NA16NOS4780203.”

6. We note that Figure 2 in your submission contain [map/satellite] images which may be copyrighted. All PLOS content is published under the Creative Commons Attribution License (CC BY 4.0), which means that the manuscript, images, and Supporting Information files will be freely available online, and any third party is permitted to access, download, copy, distribute, and use these materials in any way, even commercially, with proper attribution. For these reasons, we cannot publish previously copyrighted maps or satellite images created using proprietary data, such as Google software (Google Maps, Street View, and Earth). For more information, see our copyright guidelines: http://journals.plos.org/plosone/s/licenses-and-copyright.

1. You may seek permission from the original copyright holder of Figure 2 to publish the content specifically under the CC BY 4.0 license. 

Reviewers' comments:

Reviewer's Responses to Questions

**Comments to the Author**

1. Is the manuscript technically sound, and do the data support the conclusions?

Reviewer #1: Yes

Reviewer #2: Yes

2. Has the statistical analysis been performed appropriately and rigorously? 

Reviewer #1: Yes

Reviewer #2: Yes

3. Have the authors made all data underlying the findings in their manuscript fully available?

Reviewer #1: Yes

Reviewer #2: Yes

4. Is the manuscript presented in an intelligible fashion and written in standard English?

Reviewer #1: Yes

Reviewer #2: Yes

5. Review Comments to the Author

Reviewer #1: The authors derive a geospatial model for predicting hypoxic area, volume and thickness based on real-time data from multiple monitoring programs within the notorious besieged northern Gulf of Mexico hypoxic region. The model utilizes a space-time geostatistical framework and implements a rank-based inverse normal transformation to improve estimates of hypoxic thickness, an important metric typically overlooked. The authors are experts within the arena of geospatial modeling and their geostatistical methodology represents a sound alternative approach. Lack of correlation between hypoxic thickness and the other hypoxic metrics of area and volume, as well as a progressive increasing trend in thickness over the last three decades within the region underscore the importance of hypoxic thickness as an informative metric. The explicit space-time dimensions of the geospatial framework used constitute a novel aspect relative to other approaches, such as biogeochemical process models. Another strength of the model is the validation performed using actual data to identify potential biases precluding prediction accuracy. Potential environmental drivers are also evaluated. Using multiple regression to analyze output from the model, spring nitrogen loading was shown to be the primary driver of hypoxia within the system, in addition to several subsidiary drivers involving meteorological patterns. Subregions within the northern GoM sometimes responded differently to the same meteorological drivers. One of the biggest benefits of the study is the heuristic value in comparison with other approaches aimed at understanding the evolution of hypoxia within the northern Gulf of Mexico.

Reviewer #2: PONE reviewer comments Matli & Obenouer

This analysis extends prior analyses of area and volume to include thickness as a metric with which to compare annual and monthly trends and to support resource management decisions. Thickness as a metric is more difficult to understand compared to area (a synthetic number) and volume (also a synthetic number) compared to thickness, which as a synthetic number, is more difficult to understand because it is compiled from multiple hydrographic profiles across a broad range of depths and geographic locations and represents a unified metric that seems less useful as a management tool, either in reduction of nutrient inputs or fishery-related resources.

There is also a mixture of geographically different monitoring cruises, especially differences between TAMU and EPA compared to each other and to LUMCON. Additionally, LUMCONT is a one- or two-day cruise to specific transects contained within the larger LUMCON MC station locations during the shelfwide cruises. It is not clear how these various hydrographic profiles become a monthly or annual metric for statistical analyses.

Many stated results, e.g., hypoxia is more likely to occur in July and August than in June, is already well known. Winds from the east in spring disperse nutrients and fresh water from the Mississippi River in the spring season support the phytoplankton biomass that eventually falls to the sea bed and is respired.

Many general terms, e.g., For volume, the corresponding results are 0, 2, 11, 10, and 1 years, respectively, indicating that peak volume tends to lag peak area, at least on average

…..taken together, these results suggest both patterns as well as aberrations in the seasonal….

There is little doubt that a tremendous amount of consistent, strong analytical methods, and a thorough understanding of the metrics provided are correct. These are trademarks of these co-authors.

There needs to be a more concise presentation of the results and their interpretation.

Small things that caught my eye:

Inconsistent use of ….hypoxia… as a noun, and …..hypoxic…..as an adjective ……hypoxia metrics, when considered along with other riverine inputs and meteorological factors in multiple regression models. Hypoxic volume,….

L36 …..the size of the hypoxic zone

L 63, Louisiana Universities Marine Consortium

L 80, thickness of hypoxia across the Gulf hypoxia region from May, second hypoxia should be hypoxic, and so on through the manuscript.

L 101, change to ….sampling frame….

Fig. 1, a mixture of font styles

L 172, multiple cruises in each of the years analyzed, but the geographic footprints of those cruises were quite different.

L 175 et seq. Selection of ‘seasons’ does not appear to be consistent, but rather the months in which there were cruises.

L186, Mississippi and Atchafalaya rivers

L 212, a lag of 12 days

L 261, is there a reference for this statement ROMS vs FVCOM?

L 300, …too little variance…

L 336, 2000 was a drought year, little stratification

L 354 – 356, these sentences are contradictory. Overall, these results suggest that most of the variability in hypoxic volume can be attributed to changes in hypoxic area. At the same time, variability in hypoxic thickness, which is only modestly correlated to area, is also important.

L 372-373, is there a conclusion here?

Fig. 5, symbols are difficult to see.

L 388-389, after reading this paragraph is there any merit to determining hypoxia thickness?

L 447, ability to constraint uncertainty…. Change to …constrain…or chloropigments

L 449 “…..Future work in improving this model can focus on incorporating covariates that capture short term changes in system conditions….” Reviewer, of these limitations, it would take dynamic measurements of conductivity, temperature, DO, Chla or chloropigments, possibly turbidity, through the water column with some type of profiler on a buoy that could capture the hypoxic thickness. Or, some type of AUV that can change position, lower raise probes, move to a new position, over some periodicity that would capture short-term variability suitable for the models. OR, several AUVs. Any of these would be prohibitively expensive and not likely to be funded by NOAA research programs, especially for a ‘thickness’ indicator that is not as informative as area or volume, for resource management.

S—1, should the journal Environmental science and technology be capitalized here and elsewhere?

Northern Gulf of Mexico is not a specific geographic location, and should be represented as …..northern Gulf of Mexico.

(National Geophysical Data Ceter, 2001) should be ….Center….

Additional details on the development of these equations is provided in Obenour et al., (2013). Should be ….are…

SI—3 Rosette-mounted samplers, however, are on large rigs with other …suggest ‘frames’ rather than rigs…. ‘rig’ also used in manuscript text.

Figures SI—6 Summer-wide daily estimates of area, volume, and thickness with the 95% CI of area, and volume for 1997-2000 Question: why are area values daily and volume and thickness are continuous (if, I am reading this graphic correctly)?

Table S9c. Candidate variables used in Summer Average regressions (MR indicates Mississippi River, AR indicates Atchafalaya River, W indicates western shelf). Capitalize “River” If Mississippi and Atchafalaya rivers is used, ‘river’ is not capitalized as it is not a geographic location.

6. PLOS authors have the option to publish the peer review history of their article (what does this mean?). If published, this will include your full peer review and any attached files.

Reviewer #1: **Yes: **Chet F. Rakocinski

Reviewer #2: No

---

## [Author Response · Author response to Decision Letter 0]

22 Aug 2024

Response to Reviewer and Journal Comments: Trends and drivers of hypoxic thickness and volume in the northern Gulf of Mexico: 1985-2018

Dear Editor and Reviewers:

We are pleased to submit our revised manuscript "Trends and drivers of hypoxic thickness and volume in Northern Gulf of Mexico: 1985-2018". We appreciate the reviewers’ positive and helpful comments. All comments have been addressed and we feel the manuscript has been improved. Our detailed responses to reviewer comments are provided below.

Sincerely

V Rohith Reddy Matli 

Journal Suggested Corrections:

We made formatting and styling changes where relevant and needed. This includes adding the figures as separate files and changing to the reference citation formats.

All the base data and code files for model development and output processing have been uploaded to a Zenodo archive and attached as a doi with the instructions needed to run the model. (DOI: https://doi.org/zenodo.13145165)

“This work was funded by NOAA grant NA16NOS4780203.”

We made relevant changes to the manuscript as suggested and added the role of funder statement to the cover letter.

“This work was funded by NOAA grant NA16NOS4780203. This is NGOMEX contribution ###.”

“This work was funded by NOAA grant NA16NOS4780203.”

We made relevant changes to the manuscript as suggested and removed all the funding information from the submitted manuscript.

We remain committed to sharing all the data from our modelling study and have created a Dryad repository with all the raw simulations. The data has been posted at this DOI (https://doi.org/10.5061/dryad.q2bvq83tw), which will be publicly available when linked to the DOI for the accepted manuscript (i.e., once we receive a manuscript DOI, we will be able to publicly activate the Dryad DOI). A reviewer copy is available here (https://datadryad.org/stash/share/AQktYZbyXudBUESThKFbHjQD7OE86NAbjctOc8u9Ksc). 

The code required to process and summarize these simulations is provided in the Zenodo archive (described in response to comment #2).

6. We note that Figure 2 in your submission contain [map/satellite] images which may be copyrighted. All PLOS content is published under the Creative Commons Attribution License (CC BY 4.0), which means that the manuscript, images, and Supporting Information files will be freely available online, and any third party is permitted to access, download, copy, distribute, and use these materials in any way, even commercially, with proper attribution. For these reasons, we cannot publish previously copyrighted maps or satellite images created using proprietary data, such as Google software (Google Maps, Street View, and Earth). For more information, see our copyright guidelines: http://journals.plos.org/plosone/s/licenses-and-copyright.

1. You may seek permission from the original copyright holder of Figure 2 to publish the content specifically under the CC BY 4.0 license. 

The basemaps in Figure 2 of our manuscript used material from mapbox designer studio. This material is not copyrighted and mapbox allows free access for usage and sharing with proper attribution. We modified our figures to include this attribution by adding the logo and text as suggested in the guidance of static & print documentation of mapbox designer (https://docs.mapbox.com/help/dive-deeper/attribution/). 

We modified the supporting information file to make the modifications as requested and modified our in-text citations to reflect these changes.

Reviewer Comments:

Reviewer #1: The authors derive a geospatial model for predicting hypoxic area, volume and thickness based on real-time data from multiple monitoring programs within the notorious besieged northern Gulf of Mexico hypoxic region. The model utilizes a space-time geostatistical framework and implements a rank-based inverse normal transformation to improve estimates of hypoxic thickness, an important metric typically overlooked. The authors are experts within the arena of geospatial modeling and their geostatistical methodology represents a sound alternative approach. Lack of correlation between hypoxic thickness and the other hypoxic metrics of area and volume, as well as a progressive increasing trend in thickness over the last three decades within the region underscore the importance of hypoxic thickness as an informative metric. The explicit space-time dimensions of the geospatial framework used constitute a novel aspect relative to other approaches, such as biogeochemical process models. Another strength of the model is the validation performed using actual data to identify potential biases precluding prediction accuracy. Potential environmental drivers are also evaluated. Using multiple regression to analyze output from the model, spring nitrogen loading was shown to be the primary driver of hypoxia within the system, in addition to several subsidiary drivers involving meteorological patterns. Subregions within the northern GoM sometimes responded differently to the same meteorological drivers. One of the biggest benefits of the study is the heuristic value in comparison with other approaches aimed at understanding the evolution of hypoxia within the northern Gulf of Mexico.

Thank you for reviewing our manuscript and considering the major contributions of the study.

Reviewer #2: PONE reviewer comments Matli & Obenouer

This analysis extends prior analyses of area and volume to include thickness as a metric with which to compare annual and monthly trends and to support resource management decisions. Thickness as a metric is more difficult to understand compared to area (a synthetic number) and volume (also a synthetic number) compared to thickness, which as a synthetic number, is more difficult to understand because it is compiled from multiple hydrographic profiles across a broad range of depths and geographic locations and represents a unified metric that seems less useful as a management tool, either in reduction of nutrient inputs or fishery-related resources.

There is also a mixture of geographically different monitoring cruises, especially differences between TAMU and EPA compared to each other and to LUMCON. Additionally, LUMCONT is a one- or two-day cruise to specific transects contained within the larger LUMCON MC station locations during the shelfwide cruises. It is not clear how these various hydrographic profiles become a monthly or annual metric for statistical analyses.

Thank you for reviewing our manuscript and providing detailed and helpful comments. 

Regarding how the disparate data sources “become a monthly or annual metric”, this is a primary feature of the space-time modeling approach, as highlighted by reviewer #1. The approach allows us to make estimates across time and space considering both spatial and temporal correlation plus large-scale spatial (geographic/bathymetric) and temporal (seasonal trends). We made an edit to the Introduction to help clarify (Line 70-72), but additional details are in the Methods and referenced articles. Even for periods of little data collection, we can still make estimates based mostly on the large-scale trends. However, in cases of extreme data paucity, excessive uncertainty could make the estimates practically useless, as highlighted for 2016 and 2019 at lines 318-324.

Many stated results, e.g., hypoxia is more likely to occur in July and August than in June, is already well known. Winds from the east in spring disperse nutrients and fresh water from the Mississippi River in the spring season support the phytoplankton biomass that eventually falls to the sea bed and is respired.

Many general terms, e.g., For volume, the corresponding results are 0, 2, 11, 10, and 1 years, respectively, indicating that peak volume tends to lag peak area, at least on average

…..taken together, these results suggest both patterns as well as aberrations in the seasonal….

There is little doubt that a tremendous amount of consistent, strong analytical methods, and a thorough understanding of the metrics provided are correct. These are trademarks of these co-authors.

There needs to be a more concise presentation of the results and their interpretation.

We reviewed the manuscript and see limited opportunity to make it more concise without cutting out important information. While some results may be “already known” in a general sense, we believe this study provides a more comprehensive and quantitative characterization of hypoxia. 

To help address the reviewer’s comment, we have moved the comparison of our new results with those of Obenour et al., 2013 to the supporting information (now referenced at Line 317). We acknowledge that this material will not be particularly important for most readers. In addition, we have also made a few minor edits for clarity and conciseness throughout the manuscript.

Small things that caught my eye:

Inconsistent use of ….hypoxia… as a noun, and …..hypoxic…..as an adjective ……hypoxia metrics, when considered along with other riverine inputs and meteorological factors in multiple regression models. Hypoxic volume,….

In the literature, we have generally seen “hypoxic” used in front of geometric properties like “area”, “volume”, and “layer”. In most other cases, “hypoxia” is used as an attributive noun, in front of “metrics”, “modeling”, “estimates”. We edited for consistency, in addition to adopting some of your related suggestions (below).

L36 …..the size of the hypoxic zone

Corrected at Line 34.

L 63, Louisiana Universities Marine Consortium

Corrected at Line 56.

L 80, thickness of hypoxia across the Gulf hypoxia region from May, second hypoxia should be hypoxic, and so on through the manuscript.

Corrected at Line 74.

L 101, change to ….sampling frame….

Changed at Line 96 and in the supporting information.

Fig. 1, a mixture of font styles

Made changes to the figure to use consistent font throughout the figure and uploaded it as Fig1.tif

L 172, multiple cruises in each of the years analyzed, but the geographic footprints of those cruises were quite different.

Made relevant changes to Lines 165-168 to address this comment.

L 175 et seq. Selection of ‘seasons’ does not appear to be consistent, but rather the months in which there were cruises.

We use “spring” and “summer” as a shorthand, which is why we put quotes around these terms. 

L186, M

---

## [Decision Letter · Decision Letter 1]

11 Sep 2024

Trends and drivers of hypoxic thickness and volume in northern Gulf of Mexico: 1985-2018

PONE-D-24-14571R1

Dear Dr. Matli,

We’re pleased to inform you that your manuscript has been judged scientifically suitable for publication and will be formally accepted for publication once it meets all outstanding technical requirements.

Kind regards,

Abdul Azeez Pokkathappada, Ph.D.

Academic Editor

PLOS ONE

Additional Editor Comments (optional):

Dear authors, Please address the reviewer comments in the manuscript before finalizing it for submission.

Reviewers' comments:

Reviewer's Responses to Questions

**Comments to the Author**

1. If the authors have adequately addressed your comments raised in a previous round of review and you feel that this manuscript is now acceptable for publication, you may indicate that here to bypass the “Comments to the Author” section, enter your conflict of interest statement in the “Confidential to Editor” section, and submit your "Accept" recommendation.

Reviewer #2: All comments have been addressed

2. Is the manuscript technically sound, and do the data support the conclusions?

Reviewer #2: Yes

3. Has the statistical analysis been performed appropriately and rigorously? 

Reviewer #2: Yes

4. Have the authors made all data underlying the findings in their manuscript fully available?

Reviewer #2: Yes

5. Is the manuscript presented in an intelligible fashion and written in standard English?

Reviewer #2: Yes

6. Review Comments to the Author

Reviewer #2: Matli PONE Hypox thickness and volume of hypoxia nGoMx

Some thoughts came to mind as I re-read this manuscript, not particularly with the quality of the manuscript, but with some of the results. I can see why thickness is less tractable than area or volume because the thickness feature is a result of winds piling up water masses to the east at the time when Gulf coastal winds shift from the east to from the west. [Also noted in the last sentence of the abstract.] There are also summer conditions where winds from the north may prevail or from the south, or a switching in the prevailing winds during the research cruise. These several conditions would influence the volume (not so much the total) and thickness.

….on average, there is a seasonal lag in peak hypoxic volume and thickness compared to hypoxic area…..

Are there any suggestions about the formation of hypoxia beginning in the sediments with respiration of organic matter and then migrating up into the lower water column, or a combination of this along with the respiration of settled phytoplankton below the pycnocline?

Another reason that hypoxic volume may not be considered a standard metric for the nGoMx hypoxic zone is that the system is open on the westward and southern sides, or disjunct among two areas, and therefore more difficult to estimate.

Not important, and an editor will catch these, but a space before a period in L64.

L100 delete comma after et al. Same for line 134. This may be a punctuation issue for the journal. There is usually not a comma after et al. before a date. No more marked.

L217, meaning of ‘severe’ Again line 221, and elsewhere, L229, L348 Is this a relative value of volume or thickness?

Pointing out differences in the ‘east’ and ‘west’ portions of the study area is useful.

Conclusion, mixture of past and present tenses.

7. PLOS authors have the option to publish the peer review history of their article (what does this mean?). If published, this will include your full peer review and any attached files.

Reviewer #2: No

---

## [Editor Report · Acceptance letter]

29 Sep 2024

PONE-D-24-14571R1 

PLOS ONE

Dear Dr. Matli, 

I'm pleased to inform you that your manuscript has been deemed suitable for publication in PLOS ONE. Congratulations! Your manuscript is now being handed over to our production team.

Kind regards, 

on behalf of

Dr. Abdul Azeez Pokkathappada 

Academic Editor

PLOS ONE